# Triangle Search: An Anytime Beam Search

**Sofia Lemons,**[1,2] **Wheeler Ruml,**[1] **Robert C. Holte,**[3] **Carlos Linares López,**[4]

[1] University of New Hampshire
[2] Earlham College
[3] University of Alberta, Alberta Machine Intelligence Institute (Amii)
[4] Computer Science and Engineering Department, Universidad Carlos III de Madrid
sofia.lemons@earlham.edu, ruml@cs.unh.edu, rholte@ualberta.ca, carlos.linares@uc3m.es

## Abstract

Anytime heuristic search algorithms try to find a (potentially suboptimal) solution as quickly as possible and then work to find better and better solutions until an optimal solution is obtained or time is exhausted. The most widely-known anytime search algorithms are based on best-first search. In this paper, we propose a new algorithm, triangle search, that is based on beam search. Experimental results on a suite of popular search benchmarks suggest that it is competitive with fixed-width beam search and often performs better than the previous best anytime search algorithms.

## Introduction

In many applications of planning, it is convenient to have a heuristic search algorithm that can flexibly make use of however much time is available. The search can be terminated whenever desired and returns the best plan found so far. Dean and Boddy (1988) termed these *anytime* algorithms. Russell and Zilberstein (1991) further differentiated between *interruptible* algorithms, which quickly find a solution and then find better solutions as time passes, eventually finding an optimal plan if given sufficient time, and *contract* algorithms, which are informed of the termination time in advance and thus need only find a single solution before that time. Anytime algorithms have been proposed as a useful tool for building intelligent systems (Zilberstein 1996; Zilberstein and Russell 1996). While only a few contract search algorithms have been proposed (Dionne, Thayer, and Ruml 2011), interruptible algorithms have been widely investigated and applied. They have proven particularly useful in robotics applications, including self-driving cars (Likhachev and Ferguson 2009).

As we review below, the most well-known interruptible anytime heuristic search algorithms are based on best-first search. Best-first search is attractive as it is the basis for the optimally-efficient optimal search algorithm A* (Hart, Nilsson, and Raphael 1968) and it is well understood. However, because anytime algorithms are intended for use cases in which the solutions found do not need to be proven optimal, and are not even expected to be optimal, it is not obvious that best-first search is the most appropriate choice of algorithmic architecture.

In this paper, we propose an interruptible algorithm called triangle search that is inspired by beam search, which is based on breadth-first search. Triangle search is based on an incrementally widening beam search and is simple to implement. We study triangle's performance experimentally on several popular heuristic search benchmarks. We find that triangle search outperforms previously-proposed anytime search algorithms in the majority of cases tested. Furthermore, it tends to find solutions of comparable cost at similar times when compared to fixed-width beam search, implying that it can also serve as a convenient substitute for conventional beam search that just happens to be anytime.

## Background

Before presenting triangle search, we first review relevant prior work in anytime search and beam search.

### Anytime Search

Most previous anytime heuristic searches are based on weighted A* (Pohl 1973). For example, anytime weighted A* (AWA*) (Hansen and Zhou 2007) uses $f'(n) = g(n) + w \times h(n)$. It retains a current incumbent solution and continues searching for better solutions until there are no open nodes with $f(n) = g(n) + h(n) < g(incumbent)$, thus proving that the incumbent is optimal.

Anytime Repairing A* (ARA*) (Likhachev, Gordon, and Thrun 2004) also uses a weighted heuristic, but applies a schedule of decreasing weights ending with a weight of 1. When a solution is found, it decreases the weight according to the schedule and reorders the open list. It terminates after finding a solution with $w = 1$ or after exhausting all nodes with $f(n) < g(incumbent)$.

Thayer, Benton, and Helmert (2012) present the Anytime EES (AEES) algorithm which requires no weighting parameter and explicitly works to minimize the time between finding new solutions by using distance-to-go estimates $d(n)$. $d(n)$ estimates the distance to a goal state in terms of the number of state transitions, without regard for cost. Often this can simply be a unit cost heuristic. AEES maintains an open list ordered on an error-adjusted evaluation function $\hat{f}(n)$, a focal list ordered on an error-adjusted distance-to-go measurement $\hat{d}(n)$, and a cleanup list ordered on $f(n)$. It compares the current incumbent solution's cost to the lowest

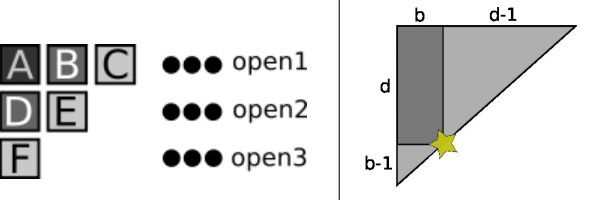

Figure 1: The exploration of triangle search (left). The overhead of triangle search compared to monobead (right).

$f$-value among open nodes to determine a bound for solution quality, $w$. The focal list maintains only nodes which have $\hat{f}(n) < w \cdot \hat{f}(b)$ where $b$ is the lowest $\hat{f}$-valued node, because these are predicted to lead to a solution that is better than the current incumbent.

## Beam Search

Beam search (Bisiani 1987) is an incomplete and suboptimal variant of breadth-first search. It expands only a fixed number of nodes at each depth level of the search, referred to as the beam width $b$. All nodes from the beam at the current level are expanded and then best $b$ among those descendants are selected to be expanded. It continues to search until either a solution is found or until no new states are reachable from the current depth. Typically node selection for beam search is based on $f(n)$ or $h(n)$, but a variant of beam search using $d(n)$ called bead outperforms beam search in non-unit cost domains (Lemons et al. 2022).

One issue with beam search is that when the beam width $b$ is increased, sometimes a lower quality solution is returned. Lemons et al. (2022) proposed the algorithms monobeam (using $f(n)$) and monobead (using $l(n) = depth(n) + d(n)$) which address this issue. These algorithms regard the beam as an ordered sequence of numbered *slots*. To fill beam slot $i$ for the next depth level, the node at slot $i$ of the current depth level's beam is expanded, its children are added to that depth level's priority queue, and the best available child is selected. This iterates for values of $i$ from 1 to $b$, with the priority queue retaining any children that were not selected to fill previous slots. The node selected for slot $i$ at depth $d$ is thus restricted to be a child of a node in slots 1 through $i$ at depth $d-1$. This careful selection order prevents children of nodes at later slots from supplanting children of earlier slots and preserves any solutions that would have been found by searches with a narrower beam.

## Triangle Search

Triangle search can be seen as an iteratively widening and deepening monobead search, conceptually resulting in a triangular shape (see Figure 1). At each iteration, it is allowed to expand one additional node from each previously used depth level, and one node from a new depth level. In the left panel of Figure 1, the node in position A is the only node expanded in the first iteration. The second iteration consists of expanding nodes in positions B and D. And the third iteration entails expanding nodes in positions C, E, and F. Open lists are maintained for each active depth level and when a node is expanded, its children are added to the open list of

---

**Algorithm 1:** Pseudocode for Triangle.

```
1  begin
2      open ← ∅
3      openlists ← [open]
4      closed ← ∅
5      incumbent ← node with g = ∞
6      create start node and add to open
7      while non-empty lists exist in openlists do
8          extend openlists with ∅ slope times
9          for i = 1 … (length(openlists) − 1) do
10             n ← remove first node from openlists[i]
11             while f(n) ≥ g(incumbent) do
12                 n ← remove first node from open
13             add n to closed
14             children ← expand(n)
15             for each child in children do
16                 if f(child) < g(incumbent) then
17                     if child is a goal then
18                         incumbent ← child
19                         report new incumbent
20                     else
21                         dup ← child's entry in closed
22                         if child not in closed or
                              g(child) < g(dup) then
23                             add child to
                                  openlists[i + 1]
24         trim empty lists from openlists
25     return incumbent
```

the depth level below. This also limits which nodes have influence on the selection for a given position. For example, at the time a node is selected for position D $open2$ would contain only children of the nodes in positions A and B. The node selected for position E, however, could be a child of nodes from positions A, B, or C. The node selected for position F must be a child of nodes from D or E, which potentially include descendants from positions A, B, and C.

In its implementation (Algorithm 1), triangle search needs only to maintain a collection of open lists (one per depth explored) ordered by distance-to-go estimates and a single closed list. So long as there are nodes left to explore, the algorithm loops over depth levels from shallowest up to (but not including) deepest (line 9), selecting a node to expand from that level's open list (line 10), re-selecting if the node has $f(n) \geq$ the incumbent solution's cost (line 12). Once the node is expanded, its children are evaluated for whether they are goals (line 17), worse quality duplicates to be discarded (line 22), or insertion into the next level's open list (line 23). When a goal state is generated which has better cost than the incumbent, it is stored as a new incumbent and reported to the user. Empty open lists are added at the beginning of the main loop (line 8), to ensure that the depth of the search can increase each time. The search terminates when there are no open nodes across all open lists, returning the incumbent solution.

In the course of searching, some of the open lists can be-

come empty. These open lists can only be filled by children of nodes in the open list for the depth above. Therefore, when the shallowest depth open lists become empty, they will never be filled again and no longer need to be iterated over for the search to proceed. Likewise, if a sequence of depths' open lists become empty at the deepest levels, none of these need to be iterated over until the open list above them has nodes in it again. Therefore, *openlists* can be made to track the highest and lowest occupied depth-levels at a given time and iterate only these active open lists (line 24).

While our discussion so far has assumed only one new depth level explored per iteration, the slope parameter (line 8) makes it possible for triangle search to explore more than one new depth level per iteration. This adjusts the algorithm's balance between deeper versus wider exploration.

## The Behavior of Triangle Search

Triangle search is complete and optimal when given an admissible heuristic. It only prunes nodes whose $f$-value is greater than or equal to the $f$-value of the incumbent, so given infinite time and memory it will search the entire reachable portion of the state space, returning a solution if one exists. By the same argument, it will eventually return an optimal solution if one exists.

Triangle search is related to monobead, in the sense that both algorithms select nodes for expansion at a given level one at a time, preventing nodes from later slots of the beam from affecting the search order in earlier slots. One difference between the search order of triangle search versus monobead is that in triangle search the children of the node expanded at the previous level can be selected at the next level, allowing the node expanded from slot $i + 1$ to influence slot $i$ at the next level. This should tend to be beneficial to triangle search because it can make a more informed selection at each level, but violates the monotonicity principles of monobead and possibly leads to a different search order. Because we do not care about monotonicity across beam widths and because bead search tends to outperform monobead (Lemons et al. 2022), we use the algorithm as presented in Algorithm 1 in the experiments below. However, for ease of theoretical analysis, we introduce a more constrained version of the triangle search algorithm that we call StrictTriangle.

The primary difference between triangle search and StrictTriangle is that a node to be expanded from the next level is selected before any children from the current level are added to the open list (moving lines 10–12 after line 14 and pre-selecting a value for $n$ before the loop at 9). This limits the selection of nodes for expansion to only nodes which came from the same slot or earlier. The slope of StrictTriangle is limited to a value of 1 (removing the loop around line 8). It also tracks the width at which a node was expanded in order to only prune duplicate nodes when they were previously encountered at the same width or less (only counting a node as duplicate at line 22 if $widthseen(dup) \leq widthseen(child)$).

StrictTriangle's overhead relative to monobead is bounded. The right panel of Figure 1 illustrates. The rectangle represents the work done by a monobead search with beam width $b$ that finds a goal as the child of a node at depth $d$, i.e. after expanding $db$ nodes (or slightly fewer if not enough nodes exist at the top of the tree). StrictTriangle search will find that goal by expanding at most $db + (d^2 - d)/2 + (b^2 - b)/2$ nodes, where the second term is the 'righthand triangle' due to unnecessary broadening at the top of the search tree and the third term is the 'lower triangle' due to unnecessary deepening at the bottom of the tree. It is possible that StrictTriangle may perform fewer expansions than this if it finds a solution that monobead would not and uses that solution's cost for pruning.

**Theorem 1.** *If monobead would discover a goal as a child of a node at depth $d$ in a particular beam slot $b$, then StrictTriangle will perform at most $(d^2 - d)/2 + (b^2 - b)/2$ more expansions to find that goal or a better one.*

*Proof.* With each increase in depth, StrictTriangle expands one more node at each previous level, if enough nodes exist at those levels. In order for there to have been $b$ nodes expanded at depth $d$, the algorithm must have explored $b - 1$ more depth levels and the current maximum depth must be $d + b - 1$. To have reached depth $d + b - 1$, the maximum allowed expansions at the top level must be $d + b - 1$. Therefore, to reach depth $d$, StrictTriangle will expand a number of nodes no greater than

$$\sum_{n=1}^{d+b-1} n = d \cdot b + \frac{d^2 - d}{2} + \frac{b^2 - b}{2}$$

Monobead will discover the given goal after $d \cdot b$ expansions. Therefore, StrictTriangle will perform at most $(d^2 - d)/2 + (b^2 - b)/2$ more expansions than monobead to find it. □

The ratio between the depth of a solution, $d$, and the beam width at which it would be found, $b$, determines how much additional work will be required by StrictTriangle, as opposed to monobead.

As we will see in our experimental evaluation, triangle search with slope=1 is not well-suited for domains in which the heuristic is very accurate and solutions are very deep (e.g., $d \gg b$). Triangle search must do $O(d^2)$ work to reach depth $d$, whereas a best-first search might only need to do $O(d)$ if the heuristic is accurate. In this sense, triangle search with slope=1 errs too much on the side of exploration in such domains. Using a larger slope may help to reach deeper regions of the search space earlier, but will still require some extraneous exploration at higher levels.

On the other hand, triangle search is not obliged to expand nodes in order of their heuristic evaluation value (be it $f$, $h$, or $d$), meaning that it does not need to expand all nodes in a heuristic 'local minimum' or 'depression' before expanding a node with a higher value. We conjecture that triangle's similarity to monotonic beam search may aid in retaining diverse nodes during the search, preventing a large local minimum from displacing all other nodes in the beam and dominating the search.

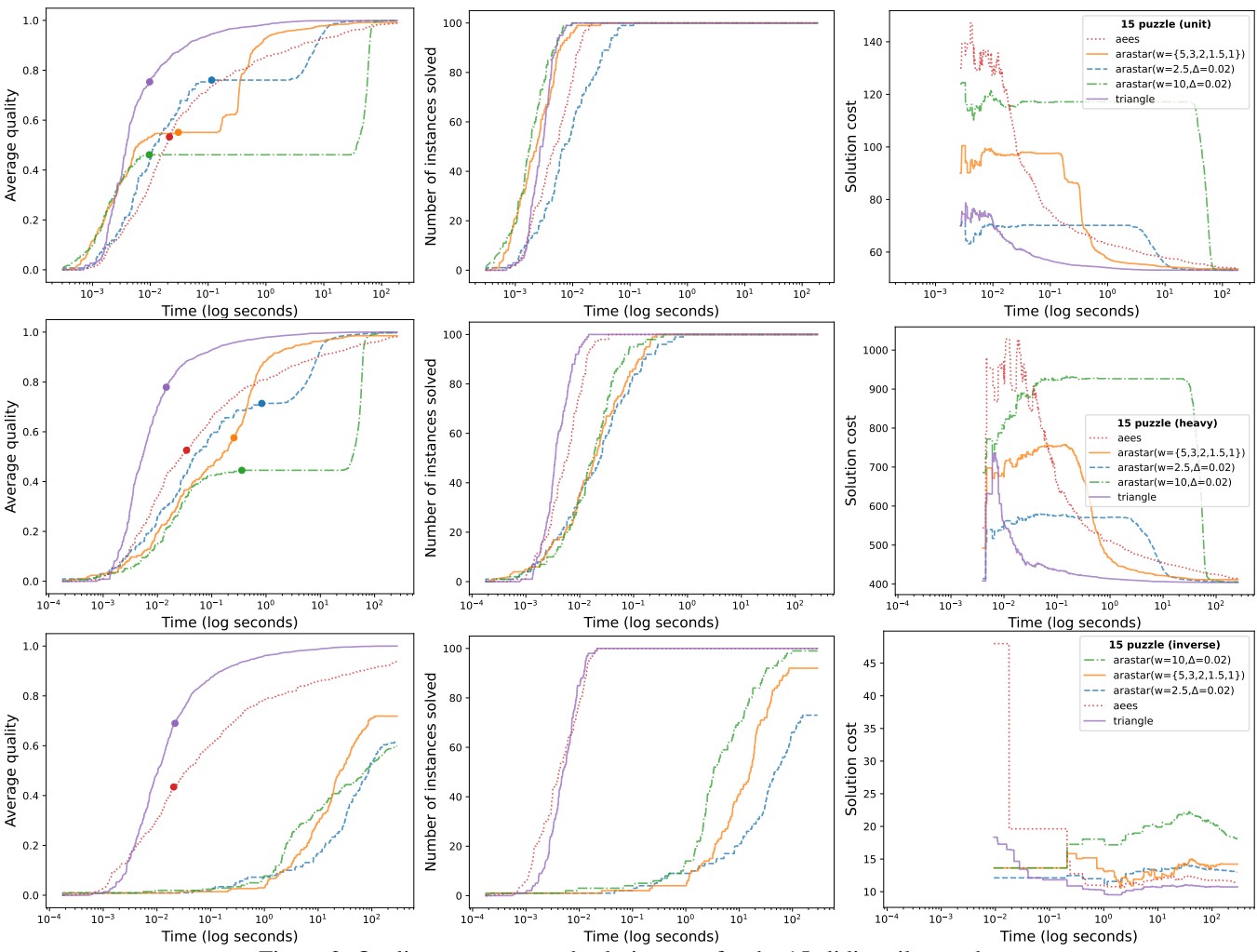

Figure 2: Quality, coverage, and solution cost for the 15 sliding tile puzzle.

## Experimental Results

We implemented triangle search and several other anytime algorithms in C++ [1] and tested their behavior on several classic search benchmarks. All algorithms were given a memory limit of 7.5GB and a time limit of 5 minutes on a 2.60 GHz Intel Xeon E5-2630v3. ARA* was tested with several configurations used in previous evaluations: initial weights of 10 and 2.5 with a decrement of 0.02 (Likhachev, Gordon, and Thrun 2004) and a weight schedule of 5, 3, 2, 1.5, 1 (Thayer, Benton, and Helmert 2012). We use a default slope of 1 for triangle unless otherwise specified.

We summarize results for these algorithms in three different ways. First, we provide quality for each algorithm averaged across all instances, where quality is defined to be the cost of the best known solution (optimal where known, otherwise the best solution ever provided by any of the tested algorithms) divided by the cost of the current incumbent solution ($\infty$ if none, giving quality 0). This allows us to average information from all instances even if an algorithm has not solved that instance yet. In the quality plots, a dot marks when a given algorithm has reached full coverage (at

least one solution has been found for all instances). Next, we provide coverage, which is simply the number of instances solved by a given algorithm at a particular time. Finally, we provide actual solution costs provided by each algorithm, averaged across all instances that have been solved by all displayed algorithms at that specific time.

### Sliding Tile Puzzle

Six different cost models of sliding tiles were used in our experiments: unit cost; heavy cost, where moving tile number $t$ costs $t$; sqrt cost, moving $t$ costs $\sqrt{t}$; inverse cost, $1/t$; reverse cost, $\#tiles - t$, and reverse inverse cost, $1/(\#tiles - t)$. We tested these cost models on the 15-puzzle (4x4) and the 24-puzzle (5x5), using a cost-weighted Manhattan distance heuristic. Results for unit, heavy, and inverse costs are shown in Figures 2 and 3, while other cost models were similar to one of these and are summarized below.

In the 15 puzzle, we see that for unit and sqrt cost all algorithms are roughly comparable in terms of coverage, and in other cost models AEES and triangle search provide the best coverage with minor differences between them. Triangle search consistently provides the best solution costs in all

---

[1]Code available at https://github.com/snlemons/search.

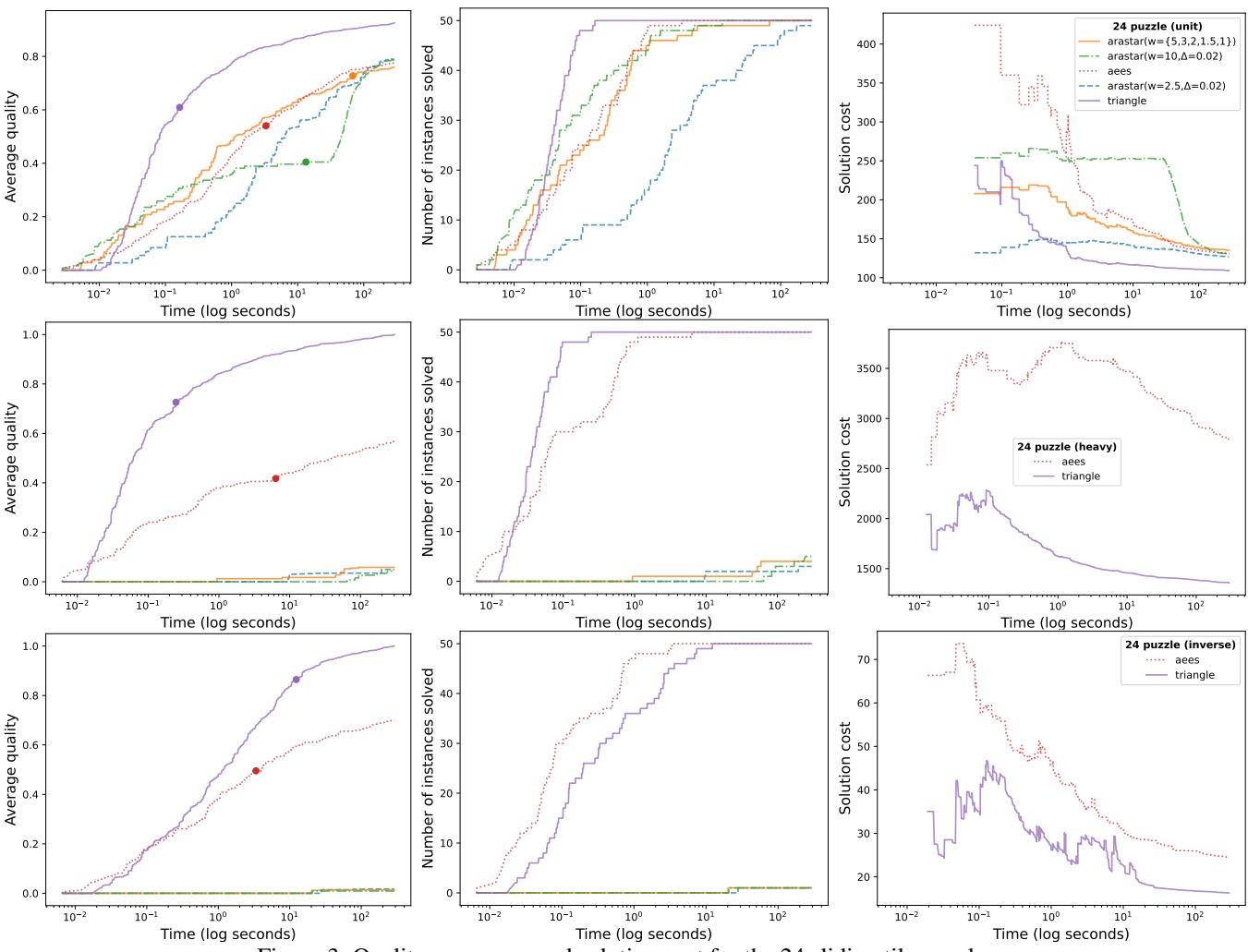

Figure 3: Quality, coverage, and solution cost for the 24 sliding tile puzzle.

cost models from 0.1 seconds onward, and the best quality at nearly all times.

In the 24 puzzle we see triangle search again providing the best quality at nearly all times. Triangle search also consistently provides the best solution costs: in unit cost triangle search dominates from around 1 second onward, and in all non unit cost models it gives the best cost at all times. In unit cost, all algorithms again provide competitive coverage at some points in time, but triangle spends a significant portion of the time with higher coverage and reaches full coverage well before any other algorithm. Triangle search also provides the best coverage for a longer period and reaches full coverage first in heavy, sqrt, and reverse inverse cost, while AEES provides better coverage in inverse and reverse cost.

## Blocks World

We tested on 100 random blocks world instances with 20 blocks. We included two different action models: 'blocks world', where blocks are directly moved to a stack as an action and 'deep blocks world', where picking up and putting down blocks each use an action, leading to longer solutions. The heuristic used was the number of blocks out of place

(any block which is not in a sequence of blocks from the table up which matches the goal state). This heuristic value is doubled for deep blocks.

For both action models, triangle search fully dominates in terms of coverage and quality at all times. In terms of solution cost, triangle search provides the best results for standard blocks world from about 0.05 seconds onward, and the best costs at nearly all times in deep blocks world.

## Pancake Problem

In the pancake puzzle, two cost models are used: unit cost, and heavy cost (Hatem and Ruml 2014), in which each pancake is given an ID number from 1 through $N$ (the number of pancakes), and the cost of a flip is the ID of the pancake above the spatula. The gap heuristic (Helmert 2010) was used, with modifications to include cost per pancake in the heavy cost model.

Triangle search with a slope of 1 takes longer than its peers to achieve coverage in the pancake problem. With 50 pancakes unit and heavy cost and 100 pancakes unit cost, at least one configuration of ARA* achieves full coverage before triangle search has solved any problems. While trian-

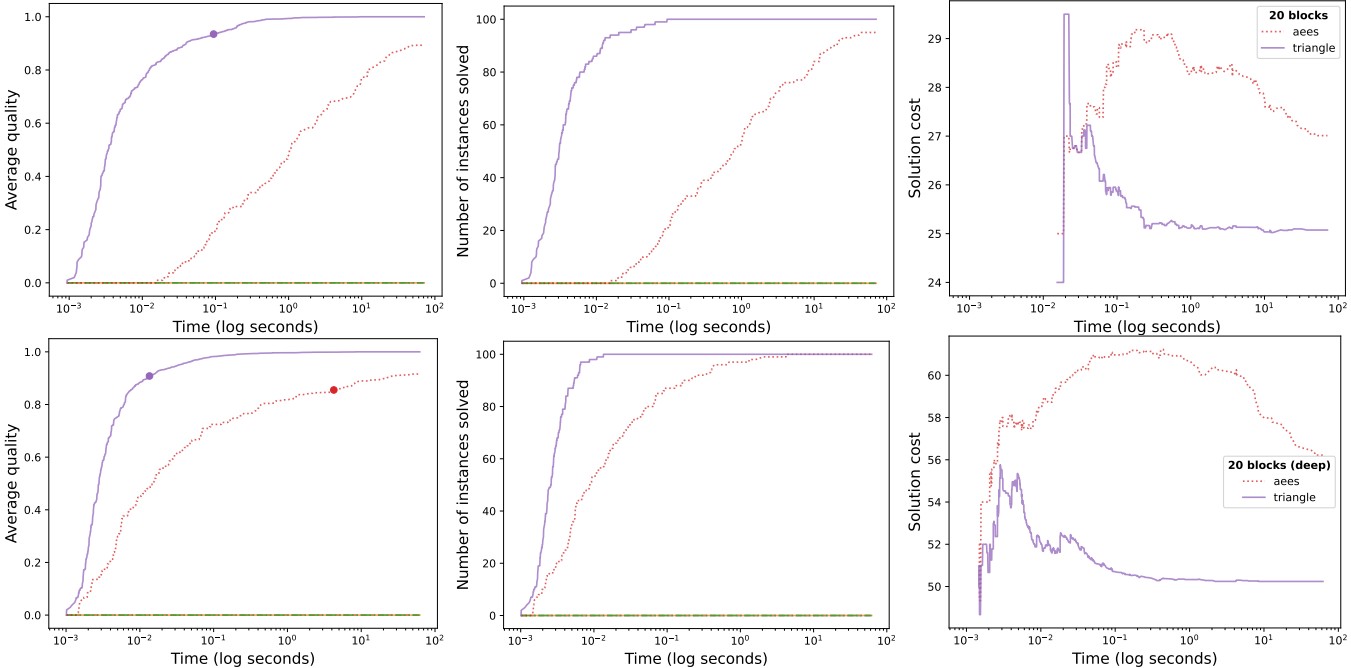

Figure 4: Quality, coverage, and solution cost for blocks world problems.

gle search does provide the best costs on instances it solves, the delay in finding a first solution is a significant downside for a user of an anytime algorithm. Only in heavy cost 100 pancake does triangle search come out best, but here it dominates in terms of quality, coverage, and cost.

We hypothesize that the poor performance of triangle search in this setting is due to the depth of solutions and the high accuracy of the heuristic. Algorithms like ARA* can focus on nodes with low $f$-values and quickly proceed to deep solutions, while triangle search must perform many expansions at higher depth levels in order to proceed deeper into the search space. This is confirmed by the success of triangle search in this setting when given a slope of 500. With a slope of 500, triangle search clearly provides the best results in both sizes and cost models of the pancake problem. It solves more instances than AEES or any configuration of ARA* at almost all times. The costs of solutions it provides are also the lowest at nearly all times.

**Vacuum World**

We also tested on the vacuum world domain (Russell and Norvig 2010), where a robot must vacuum up dirt in a grid world. In this domain we tested both the unit cost model and the heavy cost model, where the cost of movement is equal to the number of dirts which the robot has already vacuumed. The heuristic sums the number of remaining dirts, the edges of the minimum spanning tree (MST) of the Manhattan distances among the dirts, and the minimum Manhattan distance from the agent to one of the dirts. For the heavy cost model, the distance components were multiplied by the number of dirts cleaned so far. We present results for two sizes of vacuum problems: 200×200 grid with 10 dirts, and 500×500 grid with 60 dirts.

In the unit cost vacuum problems, triangle search lags dra-

matically in terms of coverage. It solves very few instances early on and reaches full coverage last out of all algorithms tested. Triangle search tends to provide the best cost once it finds solutions. Triangle search is not competitive in terms of quality until around 0.1 seconds for 200×200 problems and 1 second in 500×500 problems. Multiple configurations of ARA* give the best quality and coverage early on, as does AEES.

In the heavy cost vacuum problems, triangle search and AEES are clearly dominant. Triangle again gives the best costs among instances solved by all competitive algorithms. AEES is superior to triangle search in terms of coverage, reaching full coverage before triangle search does. As with the unit cost, triangle search does not outperform AEES in terms of quality until relatively late, but triangle ends with significantly better quality of solutions than AEES.

In both size of problems, increasing the slope seems to benefit triangle only a small amount. With slope 500, triangle search gave slight improvement in coverage and quality, but not enough to outperform the other algorithms consistently.

Of the four domains tested, the performance of triangle search is poorest in vacuum world, finding solutions later than other algorithms and only sometimes providing better solution quality. Furthermore, this is the only domain tested where solution cost tended not to decrease over time, indicating that even once solutions are found it is difficult to significantly improve upon them. We conjecture that this domain may have large plateaus in d-values. AEES is perhaps able to expand enough nodes to extend beyond these plateaus, but triangle search may re-encounter the plateaus or encounter new ones by its expansions of nodes at the shallower depth levels.

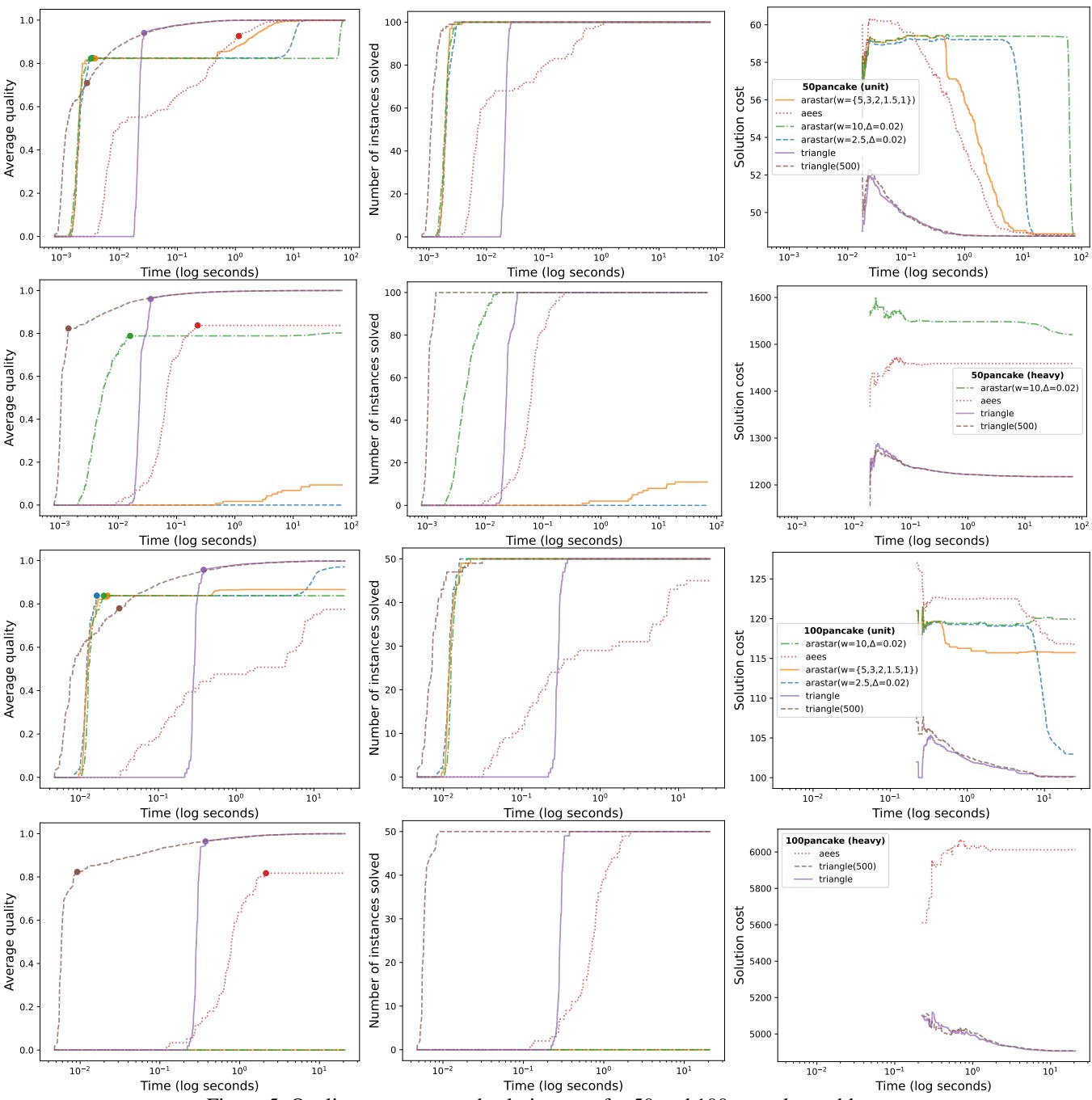

Figure 5: Quality, coverage, and solution cost for 50 and 100 pancake problem.

## Triangle Search vs Fixed-width Bead

We also performed a comparison of triangle search with fixed-width bead search (Lemons et al. 2022) to understand how the additional overhead of triangle search compares to the results obtainable by a well-selected beam width. These results can be seen in Figure 7. Because both algorithms use distance-to-go, the non-unit cost results are similar to the unit cost results in each domain, so we show only unit cost results.

In tiles and blocks world, triangle search provides competitive quality to bead at its various widths, and is able to continue improving its solution quality where bead cannot. This demonstrates that in these domains triangle search with a slope of 1 can serve as a substitute to selecting a fixed beam width.

In the 50 and 100 pancake problems, certain widths for bead outperform triangle search with slope 1 early on and triangle's improvement does not extend far above bead's. Triangle search with slope 500, however, provides superior quality than all fixed width bead searches at almost all times.

In vacuum world problems, we see that both bead and triangle struggle to find solutions overall. Few solutions are

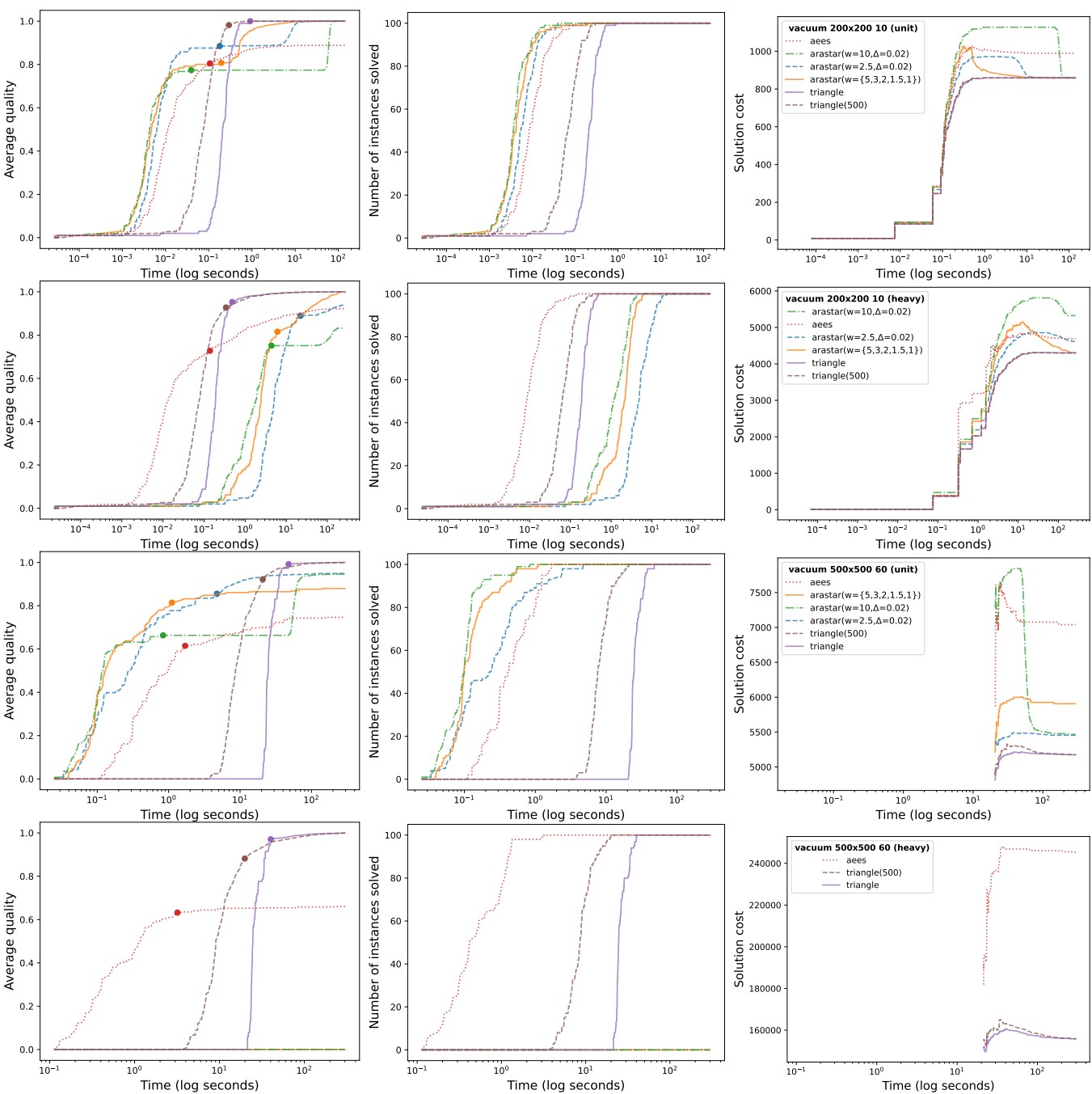

Figure 6: Quality, coverage, and solution cost for vacuum world.

found early in the time given. It appears that little improvement is gained by bead after a width of 1,000. Neither a slope of 1 or 500 give triangle search better results than most of the fixed beam widths.

## Discussion

Triangle search can be effective in settings where a wide range of beam searches are successful. If a practitioner wishes to determine whether triangle search will be useful in their problem domain, they could run beam search at several widths on a few instances. Furthermore, if low beam widths find solutions comparable to wider beam widths, this would give indication that a larger slope may be more effective for triangle.

One algorithm that is related to beam search but not evaluated here is complete anytime beam search (Zhang 1998). However, this algorithm is really an extension of depth-first search, as it only explores in a fashion similar to beam search with beam width 1 and uses backtracking in order to be complete.

BULB (Furcy and Koenig 2005) behaves like regular beam search until a given depth limit is reached, at which

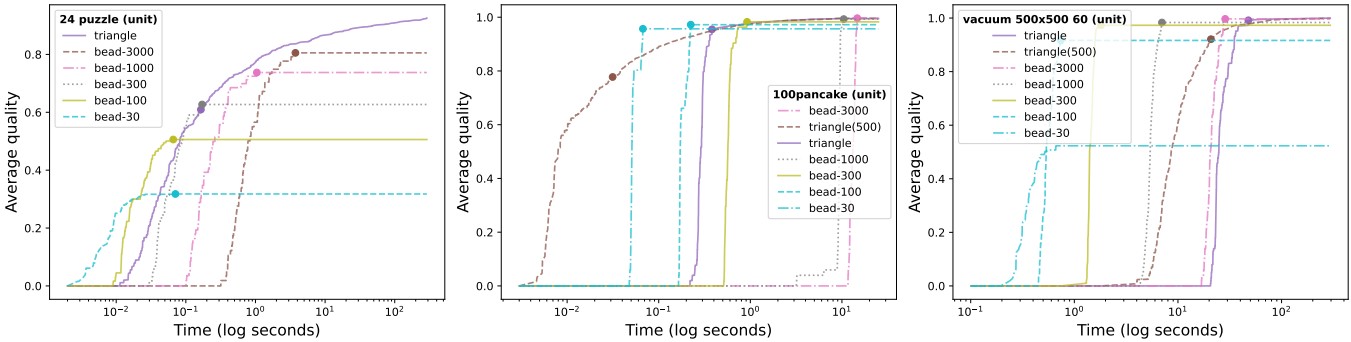

Figure 7: Quality for triangle search (anytime) versus bead (fixed-width).

point it uses backtracking to continue the search. In contrast, triangle search represents a new alternative to conventional beam search and does not require a depth limit. For huge problems where memory capacity is an issue, it would be interesting to integrate BULB-like backtracking with triangle search.

We have investigated triangle's performance with a slope of 1 and with large slopes such as 500. Additional research will be necessary to understand how to tune this parameter. Nonlinear shapes of the hypotenuse are also a possibility. This geometric investigation is superficially reminiscent of the geometric work of Chen and Sturtevant (2021) on duplicate-avoiding suboptimal search. We leave exploration of these variants to future work.

## Conclusions

Triangle search is an effective anytime algorithm with a simple design. Unlike previous anytime algorithms, which are based on best-first search, triangle is instead based on breadth-first search and it enforces exploration at a variety of depths in the search tree. In sliding tile and blocks world, it was successful with no adjustment of parameters, and in pancakes its performance was easy to improve by increasing its slope. It performed better than ARA* but was bested by AEES in vacuums, a phenomenon that deserves further study. Triangle search is often an effective replacement for fixed-width beam searches. Overall, triangle's promising performance suggests that suboptimal non-best-first heuristic search deserves further exploration.

## Acknolwedgements

We are grateful to the NSF-BSF program for support via NSF grant 2008594.

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
