# OpenReview forum: "Triangle Search: An Anytime Beam Search"
_icaps-conference.org/ICAPS/2023/Workshop/HSDIP — ICAPS HSDIP 2023_

### Official Review · Reviewer_hbV9 · 2023-04-25
**Fairly simple contribution but very relevant to HSDIP and with a supportive evaluation**

**Rating:** 6
**Confidence:** 3

**Review:**

Short Summary: The paper introduces *Triangle Search*, an optimal anytime heuristic search algorithm inspired by bead search. In essence, the algorithm extends bead search with iterative widening and deepening. An exact bound on the number of additionally expanded nodes compared to bead search is established before a goal is discovered. In the experimental evaluation, triangle search is compared against Anytime Explicit Estimation Search, anytime repairing A* and bead search on four different planning domains, where triangle search performs best in three out of four domains.

The operation of triangle search is explained well and the algorithm performs convincingly in three out of the four tested domains against the other considered anytime algorithms. However, the algorithm and the connection to bead search (Theorem 1) are very simple, which trivializes the contribution by some degree. Nevertheless, given the relevance of anytime heuristic search to the workshop and the supportive experimental evaluation, I consider this paper acceptable. There are however a few things that I found missing.

Firstly, the *slope* of the algorithm can only increase the depth of the search compared to the width, which I find to be unnecesarily restrictive. Why not introduce two parameters $d$ and $w$ so that $d$ additional layers are expanded in each iteration, and $w$ expansions are made in each layer instead of one? Then $d=2$ and $w=1$ would correspond to a slope of $2$, and $d=1$ and $w=2$ would correspond to a slope of
$\frac{1}{2}$, prioritizing the width of the search over the depth. I could imagine this being useful in domains with a high branching factor, but short solutions. Secondly, could Theorem 1 be *easily* extended to also include the slope parameter? If yes, one should include a more general claim here, considering also that the claim is fairly simple. Lastly, the experiments only consider a non-default slope parameter of 500. The evaluation could be a little more fine-grained here, although this is not a big issue.

Regarding the slope parameter: Obviously, this parameter remains fixed throughout the algorithm. Have you thought about any ways to  perhaps dynamically adapt this parameter? I suppose this is also what is hinted at by "non-linearly shaped hypotenuse" shortly before the conclusion?

Concering clarity, the paper is overall very well-written and easy to digest. A very minor exception is in Theorem 1, where the sum formula $\sum_{i=1}^{k} i = \frac{k(k+1)}{2}$ was used. The formula itself is obviously well-known, but I think it would not hurt to instantiate it in an intermediate step before simplifying to document its usage and make things more clear. Also, $slope=1$ had me confused after I read the pseudocode, since it says $0..slope$ and I expected this to be an inclusive range. A slightly bigger annoyance are the legends within the plots, in which the algorithms keep changing in order, and which are hardly transparent enough to see for example the dot for triangle search in the plot in Figure 3, left column and center row. As far as I can see (and hopefully) the line styles are always the same within a figure anyway, so a single legend per figure should suffice. The x axis labels (always time) could also be omitted and just mentioned in the caption.

Typos:
- "triangle" is written instead of "triangle search" several times (e.g., page 1 early in column 2)
- "optimally-efficient optimal" - Not really an error, but sounds really weird to me...
- page 2, col 2: "current incumbent" - "solution" is missing here
- page 5, col 2: "heavy cost 100 pancake"
- page 6, col 1: "of vacuum problem"

---

> ### Author Response · Authors · 2023-05-02
> **Very helpful and thorough**
>
> First, we would like to thank you for your time and consideration in this review. Your observations and suggestions will be very helpful in improving this paper, if accepted, as well as future work.
>
> It appears you've stated that the algorithm's simplicity reduces its significance, but please correct us if we've misunderstood. While we agree that the algorithm design is simple, we feel that this is a strength and increases triangle's importance rather than reducing it. While many of the other competitive anytime algorithms can be quite complicated and difficult to implement, triangle is both effective and straightforward.
>
> You raise an interesting point with regard to the slope parameter, which we also considered during the work. We experimented some with fractional slopes, but did not find that it offered significant improvement in any of the domains. Further exploration of this may certainly be merited.
>
> An extension of Theorem 1 to account for slope is a good idea, and we will consider it either for a final submission or future work. Thank you for this suggestion.
>
> We agree that the choice of value for slope is an important topic. The value of 500 was chosen experimentally because it gave improvement in all of the sizes of the pancake problem. We also tested a variety of slopes in vacuum and found none of them gave meaningful improvement. We can add some further mention of this in the final submission, and hope to provide more guidance and results on the proper way to set slope for the algorithm in future work.
>
> You are correct that we were implying a variable slope, among other possibilities, when mentioning "non-linearly shaped hypotenuse", and agree this could make for very interesting future work.
>
> We appreciate you pointing out the typos, the issue with the slope loop, and the suggestions around the plot legends and formula. We will work to integrate these in the final submission.

---

### Official Review · Reviewer_CmCG · 2023-04-26
**Interesting new anytime search algorithm based on beam search, exhibiting good performance in some domains, bested by ARA* in others. Experimental-heavy paper with the need of improving the presentation of the algorithm.**

**Rating:** 7
**Confidence:** 3

**Review:**

This paper presents triangle search. It maintains an open list for each
depth of the search tree and in each iteration expands a node from each
open list. A parameter called slope can be used to set the number of
depth levels that can be expanded in each iteration. The paper reports
experiments on four domains, comparing against ARA* and AEES. A
comparison against (mono?)bead search is also discussed.

The topic is clearly relevant to the workshop. The new algorithm seems
to be an interesting addition to the pool of anytime search algorithms.

While the paper is mostly well written, the description of triangle
search could be improved. For a paper that presents a new algorithm, I
found the text accompanying Algorithm 1 too unclear. The text should
guide the reader through the algorithm. What it does instead is to jump
back and forth between different parts of the algorithm, losing focus
of explaining the general idea and guiding through an execution of the
algorithm.

There is some confusion regarding slope: "for i = 0 to slope" means to
add *2* layers of open lists even when slope=1. I assume the range
should be exclusive, i.e., excluding the value slope itself? Or start
at i = 1 instead? The paper never discusses slope=0. The paper should
also explain the reasoning of the name slope, possibly better
explaining the right-hand-side of Figure 1.

I also think the pseudocode could be improved if the algorithm used a
variable depth which could be used to index openlists and to make it
easier to understand in which open list nodes are inserted. Also, this
would clarify the relation to the max-depth in each iteration, slope.

A greater problem is that the algorithm seems directly related to beam
search, and in particular monobeam and monobead search. However, the
latter two are not presented in enough detail for an non-expert to
fully understand them and in particular to judge the difference to
triangle search, making the further discussion difficult if not
impossible to understand. In particular, I never understood the
discussion of "slots" and how the width of the beam relates to
parameters (maybe slope?) of triangle search.

The presentation issue gets worse when the paper attempts, in textual
form, to describe a variant of triangle called stricttriangle and
compare this against monobead.

Finally, the amount of plots included in the paper feels overwhelming,
and their discussion is kept short and feels repetitive. Can we gain
more insights into why the algorithms behave like they do on different
benchmarks? I also wondered why the experiments didn't include a
comparison with monobead/monobeam, the closest related competitor, but
only discussed this in a separate short experiment in the end.

For a conference submission, I would require a reiteration of this
paper to address these issues, but for HSDIP this is fine.

---

> ### Author Response · Authors · 2023-05-02
> **Thank you; some responses and clarifications**
>
> Thank you sincerely for your review and the suggestions on this paper. They will be of great value to us in both a final submission of this paper, if accepted, and in future work.
>
> We agree that the discussion of the pseudocode can be improved. We will work to integrate your suggestions on this into the final submission.
>
> You are correct that the loop based on slope should be exclusive. We will clarify this in the final submission.
>
> We appreciate the guidance on clarity of the algorithm description (both triangle and strict) and adding more detail around beam and monobeam. We will add more details on these if accepted.
>
> The points you raise about the presentation of results are very helpful. We agree that gaining further insights into why the algorithms behave like they do on different benchmarks is a crucial direction. This is still ongoing work for us.
>
> A comparison against monobeam/monobead is unlikely to yield much new information, as the monotonic algorithms tend to perform slightly worse than non-monotonic beam in terms of time and solution quality because they work harder to guarantee monotonic improvement across beam widths. While we mention this briefly in the paper, we will work to clarify it in final submission and, if possible, will confirm this empirically to be able to state it more definitively.

---

### Decision · Program_Chairs · 2023-05-05

**Decision:**

Accept

**Comment:**

We are happy to announce that the paper has been accepted for the workshop, congratulations.

Both reviewers expressed their support for this decision. Please make sure to incorporate the feedback mentioned by the reviewers and fix all typos for the the final version.